 

# Disrupted in renal carcinoma 2 (DIRC2/SLC49A4) is an H⁺-driven lysosomal pyridoxine exporter

Shogo Akino, Tomoya Yasujima, Takahiro Yamashiro, Hiroaki Yuasa

**Disrupted in renal carcinoma 2 (DIRC2) has gained interest because of its association with the development of renal cancer and cosegregation with a chromosomal translocation. It is a member of the SLC49 family (SLC49A4) and is considered to be an electrogenic lysosomal metabolite transporter; however, its molecular function has not been fully defined. To perform a detailed functional analysis of human DIRC2, we used a recombinant DIRC2 protein (DIRC2-AA), in which the N-terminal dileucine motif involved in its lysosomal localization was removed by replacing with dialanine for redirected localization to the plasma membrane, exposing intralysosomal segments to the extracellular space. The DIRC2-AA mutant induced the cellular uptake of pyridoxine (vitamin B6) under acidic conditions when expressed transiently in COS-7 cells. In addition, uptake was markedly inhibited by protonophores, indicating its function through an H⁺-coupled mechanism. In separate experiments, the transient overexpression of unmodified DIRC2 (tagged with HA) in human embryonic kidney 293 cells reduced cellular pyridoxine accumulation induced by transiently introduced human thiamine transporter 2/SLC19A3 (tagged with FLAG), a plasma membrane thiamine transporter that also transports pyridoxine. The cellular accumulation of pyridoxine in Caco-2 cells as a cell model was increased by the knockdown of endogenous DIRC2. Overall, the results indicate that DIRC2 is an H⁺-driven lysosomal pyridoxine exporter. Its overexpression leads to a reduction in cellular pyridoxine accumulation associated with reduced lysosomal accumulation and, conversely, its suppression results in an increase in lysosomal and cellular pyridoxine accumulation.**

## Introduction

The gene for disrupted in renal carcinoma 2 (DIRC2) was identified as a gene that spans a recurrent breakpoint, the translocation t(2;3)(q35;q21) on chromosome 3q21, which occurs in affected members of families with hereditary renal carcinoma (Bodmer et al, 2002). A massively parallel paired-end transcriptome sequencing study described another translocation, t(2;3)(p14;q21), which is also located in the DIRC2 gene, in a prostate cancer cell line (Maher et al, 2009). These studies suggest that disruption of the DIRC2 gene plays a role in carcinogenesis. In a proteomic analysis of lysosomal membranes from human placenta, DIRC2 was identified as a putative transporter protein enriched in lysosomal membranes (Schroder et al, 2007). Accordingly, DIRC2 has been classified as a solute carrier (SLC) 49 family member, the fourth member (SLC49A4) of the family that consists of four members paralogous to the major facilitator superfamily transporters (Khan & Quigley, 2013). However, the functional characteristics of DIRC2 remain unknown, and no substrates have been identified. Of the other SLC49 family members, SLC49A1 is known as feline leukemia virus subgroup C receptor 1 (FLVCR1), a plasma membrane heme exporter that protects erythroid progenitors from heme toxicity during the heme synthesis phase of erythropoiesis (Quigley et al, 2004). SLC49A2 (FLVCR2), which is highly homologous to SLC49A1, but less so to SLC49A4, may be a plasma membrane heme importer, and it is mutated in patients with Fowler syndrome, a rare proliferative vascular disorder of the brain (Duffy et al, 2010). SLC49A3 is known as major facilitator superfamily domain containing 7 (MFSD7), which is associated with risk of ovarian cancer (Peedicayil et al, 2010), although its functional characteristics are mostly unknown.

Lysosomes are membrane-bound organelles responsible for the hydrolysis of various compounds during the turnover of small molecules primarily resulting from endocytosis, phagocytosis, and autophagy (Saftig & Klumperman, 2009). Recent studies on lysosomal enzymes have revealed that degradation in the lysosomal lumen is mediated by various hydrolases. Lysosomal hydrolases are tightly segregated with the lysosomal membrane, whereas degraded compounds are exported from the lysosomal lumen to the cytosol and can be reused to maintain cell homeostasis. The export process may be mediated, in part, by various secondary active transporters present in the lysosomal membrane. Various bioactive compounds, such as amino acids, peptides, nucleic acids, and vitamins, are exported from lysosomes by transporters (Sagne & Gasnier, 2008). Regarding the turnover of small molecules, lysosomes also play a role in cellular nutrient storage. For example, cellular storage of folate, also known as vitamin B9, is regulated by a transporter present in the lysosomal membrane for polyglutamated folate, which is an inactive retention form hydrolyzed by a lysosomal

---

Department of Biopharmaceutics, Graduate School of Pharmaceutical Sciences, Nagoya City University, Nagoya, Japan

Correspondence: yasujima@phar.nagoya-cu.ac.jp

enzyme (Barrueco et al, 1992). Functional studies of lysosomal transporters have been reported; however, the underlying molecular mechanisms have not been fully elucidated, and many of the lysosomal transporters remain unidentified.

In this study, we attempted to identify a lysosomal transporter for pyridoxine (vitamin B6). Pyridoxine is converted to the coenzyme pyridoxal-5′-phosphate and plays various physiological roles along with cytosolic enzymes, although its turnover and storage mechanism associated with lysosomes have not been elucidated. The attempt was conducted in conjunction with our recent efforts to identify pyridoxine transporters at the plasma membrane, in which we found that SLC19A2 and SLC19A3, known as thiamine transporter 1 (THTR1) and THTR2, respectively, also transport pyridoxine (Yamashiro et al, 2020). We searched bioinformatically for candidate lysosomal transporter-like proteins with unknown function to test for pyridoxine transport activity. DIRC2 was among the candidate proteins. We here describe our successful attempt to characterize DIRC2 as an H⁺-driven lysosomal pyridoxine exporter. To characterize DIRC2 function, we used a recombinant DIRC2 protein (DIRC2-AA), in which the N-terminal dileucine motif involved in its lysosomal localization was removed by replacing with dialanine by site-directed mutation for redirection to the plasma membrane (Savalas et al, 2011). The dileucine motif is located at the 14th and 15th positions with an accompanying glutamic acid at the 10th position ($E^{10}RQPL^{14}L^{15}$). As lysosomes cannot be readily used for transport experiments, the cellular model that has DIRC2-AA localized to the plasma membrane was used for examining the ability of DIRC2 to transport pyridoxine. Furthermore, we confirmed the contribution of DIRC2 to cellular pyridoxine accumulation, suggesting that DIRC2 plays a role in the turnover and cellular storage of pyridoxine.

## Results and Discussion

### DIRC2 is a lysosomal pyridoxine transporter

To examine the presumed transport function of human DIRC2, we used the DIRC2-AA mutant because several lysosomal transporters have been functionally characterized in whole cells using recombinant proteins ectopically expressed at the plasma membrane, exposing intralysosomal segments to the extracellular space (Kalatzis et al, 2001; Sagne et al, 2001; Morin et al, 2004). The advantage of this approach is to replace poorly tractable lysosomal efflux by simple whole-cell influx. We first reconfirmed the contribution of the N-terminal dileucine motif for lysosomal membrane localization in COS-7 cells by using DIRC2 fused to a FLAG tag on the N-terminus (FLAG-DIRC2) and similarly FLAG-tagged DIRC2-AA (FLAG-DIRC2-AA). In transiently transfected COS-7 cells, although FLAG-DIRC2 was primarily observed in the intracellular compartment immunocytochemically, FLAG-DIRC2-AA was predominantly observed at the plasma membrane, indicating that lysosomal targeting depends on this motif (Fig 1A).

Increased plasma membrane expression of FLAG-DIRC2-AA compared with FLAG-DIRC2 was confirmed biochemically by cell surface biotinylation using the membrane-impermeable reagent

sulfo-NHS-SS-biotin and streptavidin-agarose pull-down for the recovery of the biotinylated protein. In Western blots (Fig 1B), a small amount of FLAG-DIRC2 was also detected in the biotinylated protein fraction (plasma membrane fraction), which was also observed in an earlier study using HeLa cells (Savalas et al, 2011). However, in comparison, an increased amount of FLAG-DIRC2-AA was detected in the plasma membrane fraction, whereas the amount of FLAG-DIRC2-AA was comparable with that of FLAG-DIRC2 in total cell lysates.

An earlier study indicated that when expressed in HeLa cells, DIRC2 protein undergoes extensive proteolytic processing and resides mostly as a cleaved product at the lysosomal membrane, whereas redirected DIRC2-AA little undergoes the processing (Savalas et al, 2011). However, we found that FLAG-DIRC2-AA as well as FLAG-DIRC2 is mostly present as the cleaved product when expressed in COS-7 cells (Fig 1B). Therefore, although the reason for the difference in the proteolysis of DIRC2-AA is unknown, it was deemed possible to use DIRC2-AA to assess the function of DIRC2, which is mostly present in the cleaved form, in this study.

Using COS-7 cells transiently expressing FLAG-DIRC2-AA, we examined the uptake of pyridoxine (10 nM) under an acidic extracellular condition (pH 5.0), which mimics the natural environment for DIRC2 in the lysosome (Fig 1C). COS-7 cells were used for this assessment as a host cell with low pyridoxine uptake activity. Pyridoxine was efficiently transported in FLAG-DIRC2-AA–transfected COS-7 cells, exhibiting a fivefold greater uptake compared with that in mock cells, but not in FLAG-DIRC2–transfected COS-7 cells. This indicates that the increased pyridoxine uptake is associated with the presence of FLAG-DIRC2-AA at the plasma membrane. In an earlier study (Savalas et al, 2011), a metabolite mixture supplemented with 5% Bacto Yeast Extract (BYE) was applied to clamped oocytes expressing DIRC2-AA fused to an EGFP tag on the C-terminus (DIRC2-AA-EGFP) in an acidic extracellular medium (pH 5.0). BYE immediately induced a small, but significant, electrogenic current in DIRC2-AA–EGFP–expressing oocytes but not in water-injected or DIRC2–EGFP–expressing oocytes, indicating that DIRC2 functions as an electrogenic transporter. It appears that the BYE contained pyridoxine, and its plasma membrane transport by DIRC2-AA-EGFP induced the electrogenic current.

To further investigate the physiological role of DIRC2, we examined the expression of DIRC2 in human tissues by quantitative real-time PCR. As shown in Fig 1D, although DIRC2 mRNA was found to be ubiquitously expressed, high expression was observed in the placenta, brain, and heart. We further examined the expression in various human cell lines, in which DIRC2 mRNA was also found to be ubiquitously expressed (Fig 1E). However, it should be noted that its expression tended to be low in cell lines derived from blood cancer cells, such as HL60, HPB-ALL, HuT78, and MOLT4. An earlier study showed by Northern blot analysis that tumor cells with a t(2;3) chromosomal translocation had normal DIRC2 mRNA transcripts, indicating that the remaining intact chromosomal allele is normally transcribed, and had no additional abnormal transcripts, suggesting that disrupted transcripts of DIRC2 were absent or expressed at very low levels (Bodmer et al, 2002). Based on this, DIRC2-disrupted cells could have normal transcripts from the remaining intact allele, although the expression levels would be low because the disrupted allele cannot be transcribed. Therefore,

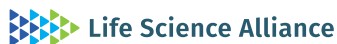

**Figure 1. DIRC2-AA mediates pyridoxine transport.**
**(A)** Immunofluorescent images showing the localization of FLAG-DIRC2 or FLAG-DIRC2-AA (green) and ATP1A1 (red) as a marker for the plasma membrane in transiently transfected COS-7 cells. Scale bar, 10 $\mu$m. **(B)** Western blots representing the protein expression levels of FLAG-DIRC2 and FLAG-DIRC2-AA in transiently transfected COS-7 cells. The blots were obtained using the total cell lysates and plasma membrane fraction (10 $\mu$g protein aliquots). The blots showing $\beta$-actin and ATP1A1 expression are for reference. **(C)** Pyridoxine uptake in COS-7 cells transiently expressing FLAG-DIRC2 or FLAG-DIRC2-AA and in mock cells. The uptake of [$^3$H]pyridoxine (10 nM) was evaluated for 1 min at pH 5.0 and 37°C. **(D, E)** Analysis of the expression of DIRC2 mRNA in various human tissues (D) and cell lines (E) by real-time PCR. **(C, D, E)** Data

taking into consideration that DIRC2 disruption can lead to carcinogenesis, there is a possibility that low DIRC2 expression in those blood cancer cell lines may be because of the presence of a disrupted DIRC2 allele, but further studies are needed to clarify the association.

### Functional characteristics of DIRC2-AA in pyridoxine transport

The pH-sensitive characteristics of pyridoxine transport mediated by DIRC2-AA were examined for a range of extracellular pHs between 5.0 and 8.0. As shown in Fig 2A, the uptake of pyridoxine (10 nM) in transiently DIRC2-AA–transfected COS-7 cells was highest at pH 5.0, at which regular uptake assays were performed. The uptake of pyridoxine decreased with increasing pH and reached a low level comparable to that in mock cells, which was low and nearly unchanged over the entire pH range, at neutral pH and above. Thus, the specific uptake of pyridoxine by DIRC2-AA was found to be highly sensitive to extracellular pH. The enhancement of the specific uptake under acidic conditions may depend on the pH of the extracellular medium, or because cells maintain their cytosol at around neutral pH, it may have resulted from an inward transmembrane $H^+$ gradient. To clarify whether DIRC2 uses an $H^+$ gradient as a driving force for transporting pyridoxine or not, we examined the specific uptake of pyridoxine by DIRC2-AA in transiently transfected COS-7 cells in the presence of protonophores, carbonylcyanide m-chlorophenylhydrazone and carbonylcyanide p-trifluoromethoxyphenylhydrazone, in the uptake solution at pH 5.0 to dissipate the $H^+$ gradient across the plasma membrane (Fig 2B). Treatment with these protonophores significantly reduced the specific uptake, indicating that an inwardly directed $H^+$ gradient is required for pyridoxine transport. This observation suggests that DIRC2 transports pyridoxine in an $H^+$-coupled manner. It should be noted, however, that pyridoxine is increasingly cationized with a decrease in pH in the pH range where the specific uptake was highly pH-sensitive because it contains a pyridine structure and a nitrogen located at the first position can be protonated with pKa 5.1 (Dos Santos et al, 2010). Therefore, these is a possibility that cationized pyridoxine is preferred by DIRC2-AA for transport and a pH-dependent change in the extent of cationization is also involved in the pH-sensitive uptake.

To determine whether other extracellular ions influence DIRC2-AA–mediated pyridoxine transport, NaCl in the uptake solution was replaced by an isotonic concentration of KCl, Na-acetate, K-acetate, and mannitol (Fig 2C). The specific uptake of pyridoxine (10 nM) by DIRC2-AA in transiently transfected COS-7 cells was almost completely abolished when $Cl^-$ was removed by replacement with acetate or mannitol, suggesting that DIRC2 requires $Cl^-$ for the efficient transport of pyridoxine. According to an earlier study (Li et al, 2019), the $Cl^-$ concentration is higher in lysosomal lumen (60–80 mM) than in the cytosol (10–40 mM). Therefore, there is a large $Cl^-$ gradient that provides favorable conditions for DIRC2 to transport pyridoxine from the lysosomal lumen to the cytosol in the intracellular environment. The characteristic

of the $H^+$ and $Cl^-$ requirement for the pyridoxine transport suggests that DIRC2 is involved in the export of lysosomal pyridoxine to the cytosol.

To delineate the kinetic characteristics of DIRC2-AA–mediated pyridoxine transport, we examined the time courses of pyridoxine uptake in transiently DIRC2-AA–transfected COS-7 cells and mock cells. As shown in Fig 2D, the uptake of pyridoxine (10 nM) increased in proportion to time up to 1 min in DIRC2-AA–transfected COS-7 cells; however, it remained very low in mock cells. Based on this, an uptake period of 1 min was set for the evaluation of pyridoxine transport across the plasma membrane in the initial uptake phase. Kinetic analysis indicated that the DIRC2-AA–specific uptake of pyridoxine was saturable with a $V_{max}$ of 6.16 nmol/min/mg protein and a $K_m$ of 522 $\mu M$ (Fig 2E). This Michaelis constant is much higher than the serum concentration of vitamin B6, which was previously reported to be 60 nM (Naurath et al, 1995). This suggests that DIRC2 is capable of transporting pyridoxine without saturation even when the concentration in lysosomes increases because of endocytosis, phagocytosis, or autophagy, although pyridoxine concentration in lysosomes has not been fully evaluated to date. In addition, the uptake of pyridoxine at a low concentration (10 nM) in DIRC2-AA–transfected COS-7 cells rapidly increased, reaching a ninefold higher level compared with that in mock cells at 4 min, when equilibrium was almost achieved (Fig 2D). Based on these results, pyridoxine transport by DIRC2 should be fully and efficiently functional in the body.

We also examined the effect of pyridoxine analogs and cationic vitamins (1 mM) on the DIRC2-AA–specific uptake of pyridoxine (10 nM) to probe the possibility that DIRC2 may also be involved in their transport (Fig 2F). Among the tested compounds, only 4-deoxypyridoxine other than pyridoxine significantly inhibited the DIRC2-AA–specific uptake, suggesting that it may also be recognized and transported by DIRC2, although 4-deoxypyridoxine is not responsible for a coenzyme function like vitamin B6. Surprisingly, pyridoxal and pyridoxamine, which are B6 vitamins, had no inhibitory effects on the DIRC2-AA–specific uptake, suggesting that they are not transport substrates of DIRC2. Notably, THTR1 and THTR2, which have recently been found to transport pyridoxine in an $H^+$-coupled manner (Yamashiro et al, 2020), also exhibited affinity for pyridoxal and pyridoxamine. This suggests that even among compounds classified as vitamin B6 because of their similar bioactivity, there are differences in disposition characteristics. It is also notable that THTR1 and THTR2 both transport thiamine and pyridoxine, whereas thiamine was suggested not to be a transport substrate of DIRC2 because of the lack of its inhibitory effect on DIRC2-AA–specific pyridoxine uptake. Similarly, nicotinamide, which is a form of vitamin B3, may not be a transport substrate because it also did not inhibit DIRC2-AA–specific pyridoxine uptake.

### DIRC2 regulates cellular pyridoxine storage

Because the results of this study suggest that DIRC2 exports lysosomal pyridoxine to the cytosol, we determined the contribution

information: Data represent the mean ± SD of three biological replicates using different preparations of cells (C) and of three technical replicates for each sample (D, E). For statistical analysis, ANOVA followed by Dunnett's test was used (C). *$P < 0.05$ compared with the control (mock).

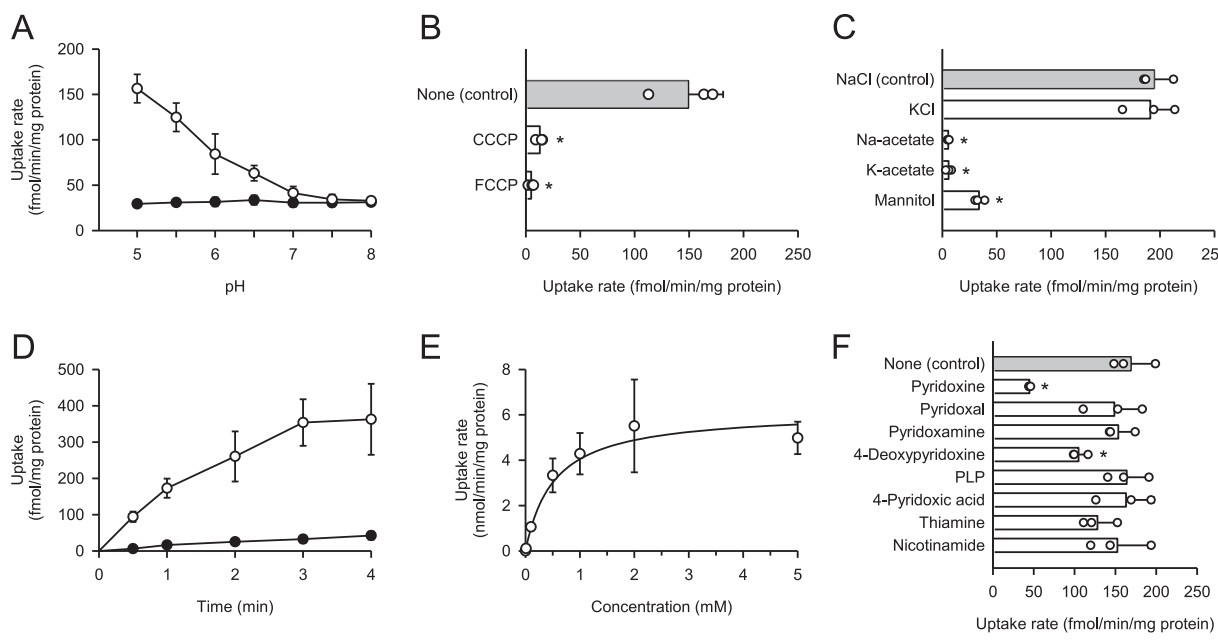

**Figure 2. Functional characteristics of DIRC2-AA transiently expressed in COS-7 cells.**
**(A)** Effect of pH on pyridoxine uptake in DIRC2-AA–transfected COS-7 cells (open circle) and mock cells (closed circle). The uptake of [$^3$H]pyridoxine (10 nM) was evaluated for 1 min at 37°C. **(B)** Effect of protonophores on pyridoxine uptake by DIRC2-AA. The specific uptake of [$^3$H]pyridoxine (10 nM) was evaluated for 1 min at pH 5.0 and 37°C in the presence of a protonophore (100 µM) or in its absence (control) after pretreatment for 5 min with or without the protonophore under the same conditions. **(C)** Effect of ionic conditions on pyridoxine uptake by DIRC2-AA. The specific uptake of [$^3$H]pyridoxine (10 nM) was evaluated for 1 min at pH 5.0 and 37°C. NaCl in the uptake solution for the control was replaced as indicated. **(D)** Time course of pyridoxine uptake in DIRC2-AA–transfected COS-7 cells (open circle) and mock cells (closed circle). The uptake of [$^3$H]pyridoxine (10 nM) was evaluated at pH 5.0 and 37°C. **(E)** Concentration-dependent uptake of pyridoxine by DIRC2-AA. The specific uptake of [$^3$H]pyridoxine was evaluated at various concentrations for 1 min at pH 5.0 and 37°C. The estimated values of $V_{max}$ and $K_m$ were 6.16 ± 0.75 nmol/min/mg protein and 522 ± 63 µM, respectively. **(F)** Effect of pyridoxine analogs and cationic vitamins on pyridoxine uptake by DIRC2-AA. The specific uptake of [$^3$H]pyridoxine (10 nM) was evaluated for 1 min at pH 5.0 and 37°C in the presence of a test compound (1 mM) or in its absence (control). PLP, pyridoxal-5′-phosphate. **(B, F)** Data information: Data represent the mean ± SD of three biological replicates using different preparations of cells (B, C, F). For statistical analysis, ANOVA followed by Dunnett's test was used (B, C, F). *$P < 0.05$ compared with the control.

of DIRC2 to cellular pyridoxine storage. As shown in Fig 3A, DIRC2 fused to an HA tag on the N-terminus (HA-DIRC2) was transiently overexpressed in human embryonic kidney 293 (HEK293) cells and the effect on the cellular accumulation of pyridoxine (10 nM) was examined after 30 min of incubation at pH 5.0. HEK293 cells were used as a human cell line with moderate DIRC2 expression (Fig 1E), in which the putative lysosomal storage system for pyridoxine could be in operation and introduced HA-DIRC2 could have an impact on that. However, transient expression of HA-DIRC2 alone did not alter the cellular accumulation, compared with that in mock cells. The unchanged cellular accumulation may have resulted from the low cellular uptake of pyridoxine because of the absence or minimal expression of endogenous plasma membrane transporters for pyridoxine in HEK293 cells. To investigate the role of lysosomal DIRC2 in pyridoxine storage, the cellular uptake of pyridoxine was increased by transient introduction of human THTR2 fused to a FLAG tag on the N-terminus (FLAG-THTR2). THTR2 is an aforementioned H$^+$-driven pyridoxine transporter, which operates at the plasma membrane. Although the cellular accumulation was increased by the introduction of FLAG-THTR2, HA-DIRC2 significantly reduced the cellular accumulation by its coexpression. The protein level of FLAG-THTR2 assessed by Western blotting was unchanged following coexpression with HA-DIRC2 (Fig 3B), and, in addition, FLAG-THTR2 was almost exclusively observed at the plasma

membrane immunocytochemically, apparently separated from HA-DIRC2 localized in the intracellular compartment (Fig 3C). Therefore, the decrease in the cellular accumulation was suggested not to be caused by a decrease in the expression or plasma membrane localization of FLAG-THTR2. It could be a result from an increase in the lysosomal export by the introduction of HA-DIRC2, which reduces the lysosomal accumulation. HA-DIRC2 was also confirmed to be mostly present as the cleaved product when expressed in HEK293 cells (Fig 3B). These results indicate that DIRC2 regulates pyridoxine storage by operating at the lysosomal membrane.

The overexpression of HA-DIRC2 at the lysosomal membrane significantly decreased the cellular accumulation of pyridoxine induced by FLAG-THTR2, as described above. To support a physiological role of DIRC2 in cellular pyridoxine storage, we compared the values of the $V_{max}/K_m$, which represents the index of transport activity, by kinetic analysis using COS-7 cells transiently expressing N-terminal EGFP-tagged DIRC2-AA (EGFP-DIRC2-AA) and THTR2 (EGFP-THTR2). Kinetic analysis indicated that the EGFP-DIRC2-AA–specific uptake of pyridoxine was saturable with a $V_{max}$ of 10.0 nmol/min/mg protein and a $K_m$ of 476 µM, resulting in a $V_{max}/K_m$ of 21.1 µl/min/mg protein (Fig 3D), and the EGFP-THTR2–specific uptake was so with a $V_{max}$ of 414 pmol/min/mg protein and a $K_m$ of 20.0 µM, resulting in a $V_{max}/K_m$ of 20.7 µl/min/mg protein (Fig 3E).

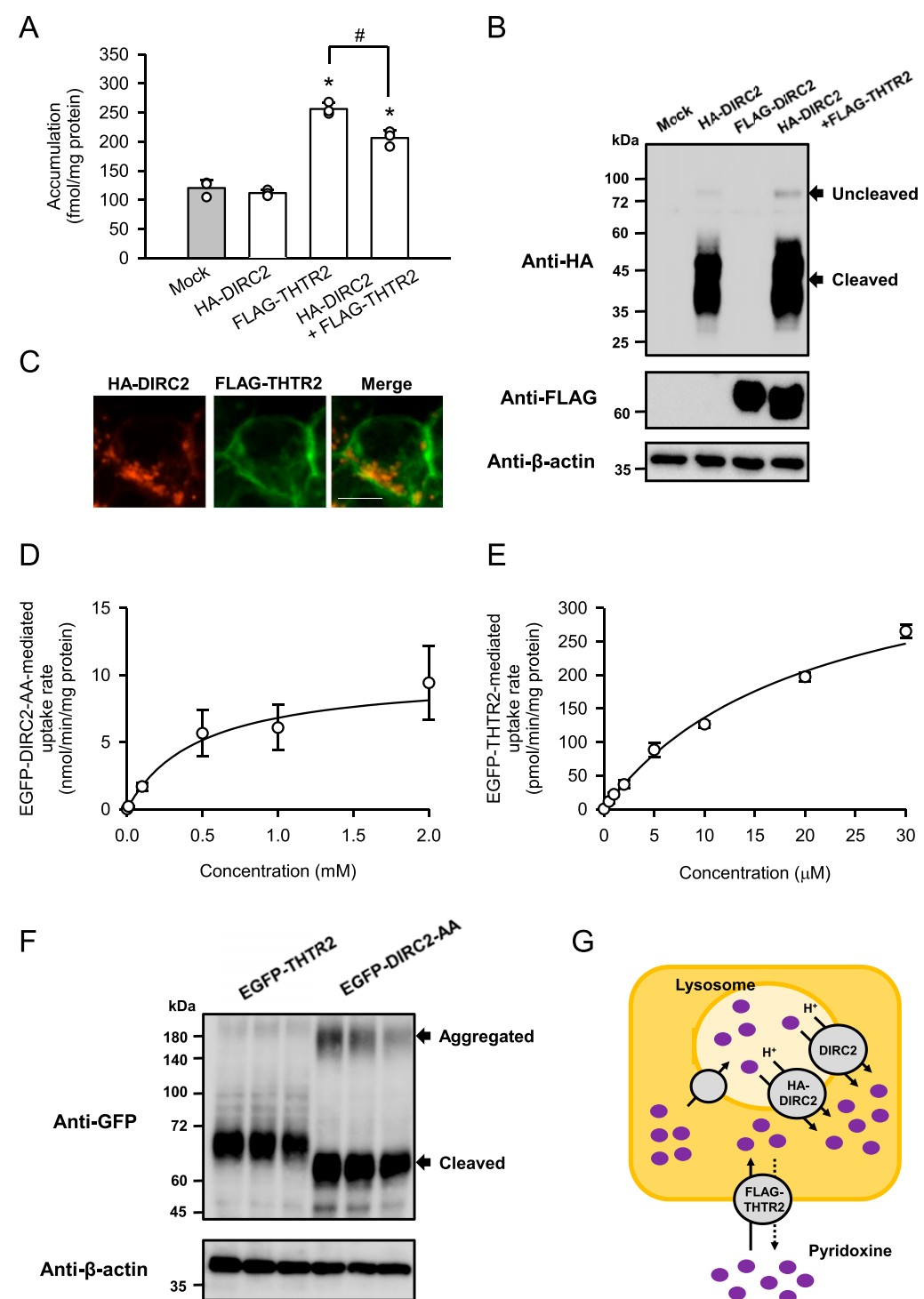

**Figure 3. HA-DIRC2 overexpression decreases cellular pyridoxine accumulation in HEK293 cells.**
**(A)** Effect of the expression of HA-DIRC2 on FLAG-THTR2–induced pyridoxine accumulation in transiently transfected HEK293 cells. The accumulation of [$^3$H]pyridoxine (10 nM) was evaluated after 30 min of incubation at pH 5.0 and 37°C in HEK293 cells, in which the plasmid for HA-DIRC2 (500 ng) was transfected with that for FLAG-THTR2 (500 ng). For expression of HA-DIRC2 or FLAG-THTR2 alone, cells were transfected with the plasmid for the designated transporter (500 ng) and empty vector (500 ng). **(B)** Western blots representing the protein expression levels of HA-DIRC2 and FLAG-THTR2 in transiently transfected HEK293 cells. The blots were obtained using the total cell lysates (10 $\mu$g protein aliquots). The blots showing $\beta$-actin expression are for reference. **(C)** Immunofluorescent images showing the localization of HA-DIRC2 (red) and FLAG-THTR2 (green) in transiently transfected HEK293 cells. Scale bar, 10 $\mu$m. **(D, E)** Concentration-dependent uptake of pyridoxine by EGFP-DIRC2-AA (D) and EGFP-THTR2 (E) transiently expressed in COS-7 cells. The specific uptake of [$^3$H]pyridoxine was evaluated at various concentrations for 1 min at pH 5.0 and 37°C. The estimated values of $V_{max}$ and $K_m$ were 10.0 ± 1.7 nmol/min/mg protein and 476 ± 19 $\mu$M, respectively, for FLAG-DIRC2-AA and 414 ± 17 pmol/min/mg protein and 20 ± 1 $\mu$M, respectively, for EGFP-THTR2. **(F)** Western blots representing the protein expression levels of EGFP-DIRC2 and EGFP-THTR2 in transiently transfected HEK293 cells. The blots (three biological

Thus, the $V_{max}/K_m$ values were comparable, indicating that the pyridoxine transport activity of EGFP-DIRC2 and EGFP-THTR2 was almost the same. A Western blot analysis of the expression of EGFP-DIRC2-AA and EGFP-THTR2 indicated that their protein expression levels were also comparable (Fig 3F). Therefore, it may be possible that a decrease in the lysosomal accumulation of pyridoxine owing to the addition of HA-DIRC2–mediated export has an impact on the THTR2-induced cellular accumulation (total amount in the cells) and results in its significant decrease (Fig 3G), as demonstrated in Fig 3A. EGFP-DIRC2-AA was also confirmed to be mostly present as the cleaved product when expressed in COS-7 cells, although there were minor Western blot bands that may represent aggregated EGFP-DIRC2-AA (Fig 3F).

To gain further evidence for the involvement of DIRC2 in cellular pyridoxine storage, we determined the effect of DIRC2 knockdown on pyridoxine accumulation in Caco-2 cells, which exhibited the highest DIRC2 expression among the human cell lines (Fig 1E) and, hence, could be highly responsive to DIRC2 knockdown. To measure cellular pyridoxine storage, the cellular accumulation of pyridoxine (10 nM) was determined after 30 min of incubation at pH 5.0. The acidic condition was set in accordance with our previous finding that the uptake of pyridoxine in Caco-2 cells occurs by $H^+$-driven transport that involves THTR1 and THTR2 (Yamashiro et al, 2020). As a result, the cellular accumulation was significantly increased in Caco-2 cells in which the expression of endogenous DIRC2 was knocked down by RNA interference (DIRC2-KD Caco-2 cells) compared with negative control cells (Fig 4A). The localization of HA-DIRC2 was also observed immunocytochemically in the intracellular compartment of transiently transfected Caco-2 cells (Fig 4B), indicating the lysosomal localization of DIRC2. Furthermore, the increase in the cellular accumulation of pyridoxine by DIRC2 knockdown was demonstrated to be associated with an increase in the lysosomal accumulation of pyridoxine (Fig 4C), which was evaluated by fractionating lysosomes after the incubation of Caco-2 cells for pyridoxine accumulation. The lysosomal fraction was confirmed to be enriched with lysosomal-associated membrane protein 1 (LAMP1), a lysosomal marker protein, whereas its level was low in the total cell lysate (Fig 4D). To check the specificity of DIRC2 knockdown, we measured the expression levels of the mRNAs of DIRC2 (Fig 4E), THTR1 (Fig 4F), and THTR2 (Fig 4G) by real-time PCR. In DIRC2-KD Caco-2 cells, the expression of DIRC2 was significantly decreased, whereas that of THTR1 and THTR2 was not. Taken together, the increase in the cellular accumulation by DIRC2 knockdown could be a result from the suppression of the DIRC2–mediated export at the lysosomal membrane (Fig 4H), which increases the lysosomal accumulation.

### Concluding remarks

This study represents an important contribution to understanding the molecular mechanism that involves DIRC2 as a newly identified $H^+$-driven lysosomal pyridoxine exporter and especially with respect to the turnover of pyridoxine, which has not been fully explored. A decrease in the function or expression of DIRC2 may cause impaired pyridoxine turnover, which reduces cytosolically available pyridoxine, with retaining pyridoxine excessively in lysosomes. This could be a causative factor for renal carcinogenesis known to be linked to DIRC2 disruption (Bodmer et al, 2002). It is also notable that vitamin B6 preparations containing pyridoxine are reportedly effective for suppressing and preventing various cancers, such as colorectal cancer (Schernhammer et al, 2008), lung cancer (Johansson et al, 2010), and breast cancer (Zhang et al, 2003). Therefore, these is a possibility that a decrease in the supply of vitamin B6, which might be caused in various organs by the impairment of ubiquitously expressed DIRC2, could also be a risk factor for the onset and development of such cancers. More importantly, the ubiquitous presence of DIRC2 in various organs may suggest that DIRC2 could potentially be involved in the regulation of the disposition of pyridoxine as a vitamin widely in the body. Although the role of the suggested lysosomal pyridoxine storage system, including the involvement of DIRC2, in the physiological processes that require pyridoxine remains to be verified, the newly identified function of DIRC2 as a lysosomal pyridoxine exporter should help guiding future studies for verification of the role and exploration of its clinical relevance. This is the first report to examine the molecular mechanism that controls the cellular accumulation and reuse of vitamins, and it will contribute to an understanding of the relationship between vitamin dynamics and certain diseases.

## Materials and Methods

### Materials

[³H]Pyridoxine (20 Ci/mmol) was obtained from American Radiolabeled Chemicals. Unlabeled pyridoxine was obtained from Tokyo Chemical Industry. DMEM was obtained from Wako Pure Chemical Industries, and FBS was from Invitrogen. All other regents were of analytical grade and commercially obtained.

### Cell culture

COS-7, Caco-2, and BeWo cells were obtained from the RIKEN BioResource Research Center, and other cells were obtained from the Cell Resource Center for Biomedical Research, Tohoku University. The cells were maintained at 37°C in a 5% $CO_2$ atmosphere in DMEM supplemented with 10% FBS, 100 U/ml penicillin, and 100 μg/ml streptomycin as described previously (Mimura et al, 2017).

### Preparation of plasmids

The cDNA for human DIRC2 (GenBank accession number, NM_032839.3) was cloned using an RT–PCR–based method as described previously (Mimura et al, 2017). Briefly, an RT reaction was

replicates) were obtained using the total cell lysates (10 μg protein aliquots). The blots showing β-actin expression are for reference. **(G)** Schematic model showing HA-DIRC2 contributing to cellular pyridoxine accumulation in HEK293 cells transiently coexpressing HA-DIRC2 and FLAG-THTR2 by exporting pyridoxine at the lysosomal membrane. **(A, D, E)** Data information: Data represent the mean ± SD of three biological replicates using different preparations of cells (A, D, E). For statistical analysis, ANOVA followed by a Bonferroni test was used (A). *$P < 0.05$ compared with the control (mock) and HA-DIRC2 alone. #$P < 0.05$.

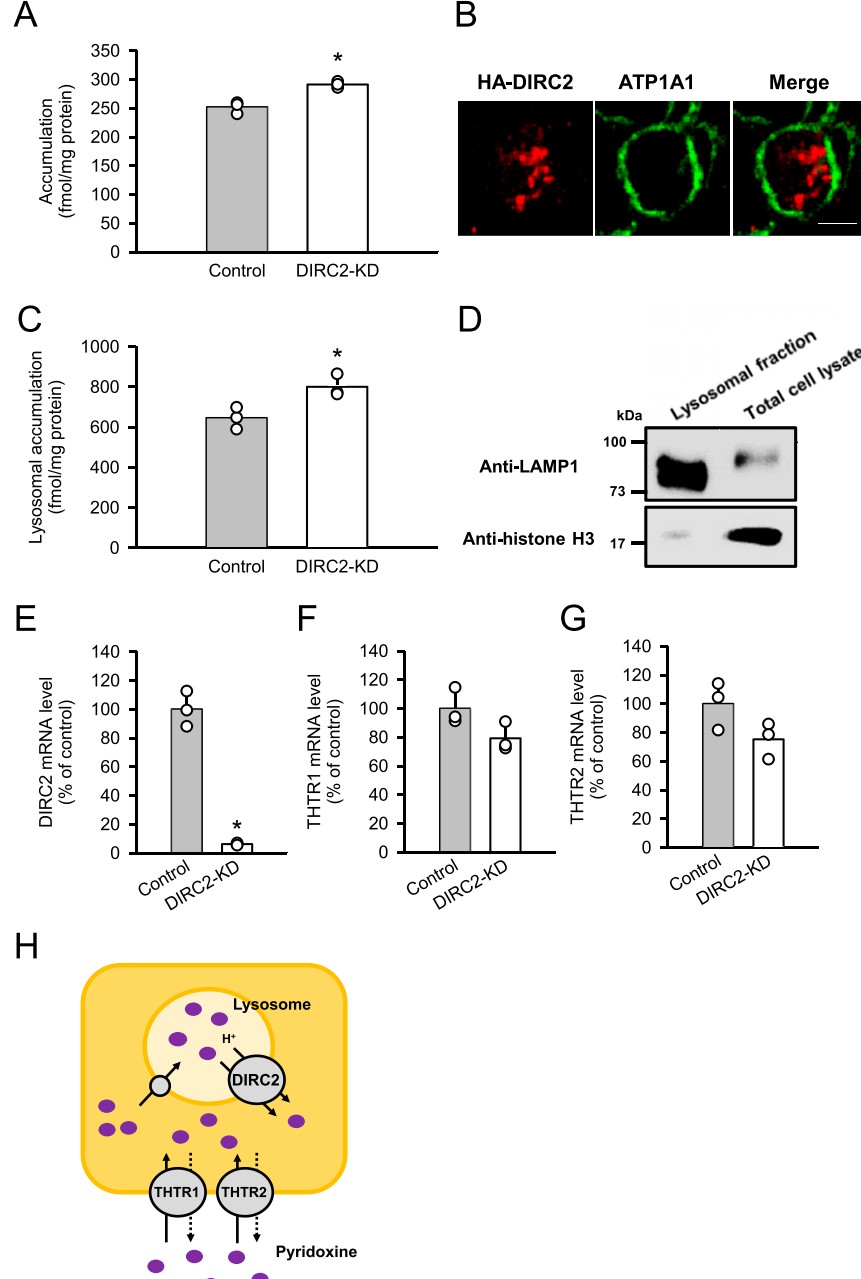

**Figure 4. DIRC2 knockdown increases cellular pyridoxine accumulation in Caco-2 cells.**
**(A)** Effect of the knockdown of endogenous DIRC2 on pyridoxine accumulation in Caco-2 cells. Caco-2 cells were cultured for 5 d after transfection with the siRNA for DIRC2 (DIRC2-KD cells). The accumulation of [$^3$H] pyridoxine (10 nM) was evaluated after 30 min of incubation at pH 5.0 and 37°C. **(B)** Immunofluorescent images showing the localization of HA-DIRC2 (red) and ATP1A1 (green) as a marker for the plasma membrane in transiently transfected Caco-2 cells. Scale bar, 10 $\mu$m. **(C)** Effect of the knockdown of endogenous DIRC2 on lysosomal pyridoxine accumulation in Caco-2 cells. The lysosomal accumulation of [$^3$H]pyridoxine was evaluated after 30 min of incubation of control and DIRC2-KD Caco-2 cells with [$^3$H]pyridoxine (10 nM) at pH 5.0 and 37°C. **(D)** Western blots representing the protein levels of LAMP1, a lysosomal marker protein, and histone H3, a nucleic marker protein, in samples (10 $\mu$g protein aliquots) of the lysosomal fraction and total cell lysate prepared from DIRC2-KD Caco-2 cells. **(E, F, G)** The levels of DIRC2 (E), THTR1 (F), and THTR2 (G) mRNA measured by real-time PCR in control and DIRC2-KD Caco-2 cells. **(H)** Schematic model showing DIRC2 contributing to cellular pyridoxine accumulation in Caco-2 cells by exporting pyridoxine at the lysosomal membrane. **(A, C, E, F, G)** Data information: Data represent the mean ± SD of three biological replicates using different preparations of cells (A, C, E, F, G). For statistical analysis, a $t$ test was used (A, C, E, F, G). *$P < 0.05$ compared with the control.

carried out to obtain a cDNA mixture from total human placental RNA (BioChain Institute) using 1 $\mu$g of total RNA, an oligo(dT) primer, and the ReverTra Ace reverse transcriptase (Toyobo). The cDNA for DIRC2 was amplified by PCR using PrimeSTAR Max DNA Polymerase (Takara Bio). A second PCR reaction was performed using the amplified product as a template to incorporate restriction sites. The PCR primers are listed in Table S1. The cDNA for DIRC2-AA was generated by site-directed mutagenesis using a PrimeSTAR Mutagenesis Basal Kit (Takara Bio) and the KOD ONE polymerase. The PCR primers for this step are listed in Table S2. The cDNA for human THTR2 (GenBank accession number, NM_006996.3) was prepared as

described in our previous study (Yamashiro et al, 2020). All final cDNA products were incorporated into the pCI-neo vector (Promega) to prepare plasmids for transfection, and their sequences were determined with an automated sequencer (ABI PRISM 3130; Applied Biosystems), as described previously (Yamashiro et al, 2019).

The plasmids for HA- or FLAG-tagged transporters were generated by transferring their coding regions into the pCI-neo vector that was modified to fuse the HA or FLAG tag to the N-terminus. For DIRC2-AA and THTR2, plasmids using the pEGFP-C1 vector (Promega) were similarly prepared for the generation of EGFP-tagged transporters.

## Preparation of transiently transfected COS-7, HEK293, and Caco-2 cells

For uptake assays and Western blot analyses, COS-7 and HEK293 cells (4.0 × 10^5 cells/ml, 0.5 ml/well) were seeded into 24-well plates coated with poly-L-lysine, transfected with the plasmid for a designated transporter (1 μg/well) using 5 and 2 μg/well of poly-ethyleneimine (Polyscience), respectively, as a transfection re-agent, unless otherwise indicated, and cultured for 48 h. Mock cells were prepared by the same procedures using empty vector. For immunofluorescence staining, COS-7, HEK293, and Caco-2 cells (3.0 × 10^5 cells/ml, 2 ml/dish) were seeded into 35-mm glass bottom dishes. COS-7 and Caco-2 cells were transfected with the plasmid for a designated transporter (4 μg/dish), and HEK293 cells were transfected with those for HA-DIRC2 (2 μg/dish) and FLAG-THTR2 (2 μg/dish) together, using 20 μg/dish of polyethyleneimine.

## Knockdown of DIRC2 in Caco-2 cells

In experiments to evaluate cellular accumulation of pyridoxine, Caco-2 cells (1.0 × 10^4 cells/ml, 0.5 ml/well) were cultured in 24-well coated plates for 6 h, transfected with 20 pmol/well of Silencer Select siRNA specific to DIRC2 mRNA (Thermo Fisher Scientific) using 1.5 ml/well of Lipofectamine RNAi MAX (Thermo Fisher Scientific), and cultured for 5 d. In experiments to evaluate lysosomal accumulation of pyridoxine, Caco-2 cells (1.0 × 10^4 cells/ml, 2.0 ml/well) were cultured in six-well coated plates for 6 h, transfected with 80 pmol/well of Silencer Select siRNA specific to DIRC2 mRNA using 6.0 ml/well of Lipofectamine RNAi MAX, and cultured for 5 d. The sequences of the siRNAs are listed in Table S3. For control, negative control DsiRNA (Integrated DNA Technologies) was used.

## Quantitation of DIRC2 mRNA by real-time PCR

Total RNA samples from various human tissues (BioChain Institute) and those from human cell lines, which were isolated by a guanidine isothiocyanate extraction method (Chomczynski & Sacchi, 1987), were used to obtain cDNA using the ReverTra Ace reverse transcriptase. Real-time quantitative PCR was done using a Luna Universal qPCR Master Mix (New England Biolabs) on a CFX Connect Real-Time PCR Detection System (Bio-Rad Laboratories) with gene-specific primers (Table S4). The mRNA expression levels were normalized to that of ubiquitin C for tissues and GAPDH for cell lines.

## Isolation of lysosomal fraction

The lysosomal fraction was isolated by a method reported previously (Hrikumar et al, 1989). Briefly, Caco-2 cells cultured in six-well plates were collected and homogenized in ice-cold homogenization solution containing 50 mM mannitol and 20 mM Hepes (pH 7.4). The homogenized sample was centrifuged at 1,500g for 10 min to remove cellular components such as nuclei, cytoskeleton, and mitochondria. The supernatant was centrifuged at 24,000g for 30 min to obtain the lysosomal fraction as the precipitate. All procedures were performed at 4°C.

## Western blot analysis

Western blot analysis was done to examine protein expression in transiently transfected COS-7 and HEK293 cells cultured in 24-well coated plates. To obtain the total cell lysate, the cells were lysed with ice-cold lysis buffer 1 (50 mM Tris–HCl, 1% SDS, 4 M urea, 1 mM EDTA, 150 mM NaCl, pH 8.0). To collect the plasma membrane fraction, transiently transfected COS-7 cells were washed twice with ice-cold Hanks' solution, added with 10 ml ice-cold Hanks' solution containing 2.5 mg of Biotin-SS-Sulfo-Osu (Dojindo), and shaken on ice for 30 min. The solution was removed, ice-cold quench buffer (50 mM Tris–HCl, 0.1 mM EDTA, 150 mM NaCl, pH 8.0) was added to the cells, and the sample was shaken on ice for 10 min. The buffer was removed, and the cells were washed once with ice-cold Hanks' solution. The cells were solubilized with 1 ml of lysis buffer 2 (1% Triton X-100, 25 mM Tris–HCl, 100 mM NaCl, pH 8.0) and gently scraped with a scraper. The whole lysate was collected in a tube, ultrasonicated, and centrifuged at 16,000g for 15 min at 4°C. The supernatant was transferred to a new tube, and Avidin-Agarose from egg white (Sigma-Aldrich) was added, and the mixture was vortexed with rotation overnight at 4°C. The sample was centrifuged at 220g for 3 min at 4°C, and the supernatant was removed, and the residual was added with 0.5 ml of lysis buffer 2. This step was repeated twice more, and the supernatant was completely removed following centrifugation at 220g for 3 min at 4°C. Finally, lysis buffer 1 was added to obtain a solubilized sample representing the plasma membrane fraction.

For examination of LAMP1 and histone H3 in Caco-2 cells cultured in six-well coated plates and treated for DIRC2 knockdown, the cells were processed to obtain the total cell lysate similarly or to isolate the lysosomal fraction.

The total cell lysates, solubilized samples representing the plasma membrane fraction, and the lysosomal fraction were processed for protein detection as described previously (Furumiya et al, 2015). The primary antibodies for the detection of tagged transporters were mouse monoclonal anti-FLAG antibody (FUJIFILM Wako Pure Chemical), mouse monoclonal anti-HA antibody (Medical & Biological Laboratories), and mouse monoclonal anti-GFP antibody (Proteintech). Mouse monoclonal anti-β-actin antibody (Proteintech) and rabbit polyclonal anti-ATP1A1 antibody (Proteintech) were used as the primary antibody for the detection of β-actin and ATP1A1, respectively, as loading controls. The primary antibodies for the detection of LAMP1 and histone H3 were rabbit polyclonal anti-LAMP1 antibody (Proteintech) and rabbit monoclonal anti-histone H3 antibody (Cell Signaling Technology), respectively. The primary antibodies were all used at a dilution of 1:1,000. The secondary antibodies, which were conjugated to horseradish peroxidase, were goat anti-mouse IgG antibody (Jackson ImmunoResearch) and goat anti-rabbit IgG antibody (Jackson ImmunoResearch), respectively, for the mouse-derived primary antibodies and the rabbit-derived primary antibodies. They were both used at a dilution of 1:10,000. Following color development using Luminate Forte Western HRP Substrate (Merck Millipore), the protein levels were determined by enhanced chemiluminescence using a ChemiDoc Touch imaging system (Bio-Rad Laboratories).

### Immunofluorescence staining

Transiently transfected COS-7 and HEK293 cells cultured in 35-mm glass bottom dishes were washed twice with PBS and incubated for 20 min at −20°C with methanol. After washing three times with PBS, the cells were incubated for 1 h at room temperature with 1 mg/ml BSA in PBS. After removing the BSA solution, the cells were incubated for 2 h at room temperature in PBS with required primary antibodies at a dilution of 1:500. Primary antibodies were mouse monoclonal anti-FLAG antibody, rabbit polyclonal anti-HA antibody (Proteintech), and rabbit polyclonal anti-ATP1A1 antibody, respectively, for the detection of FLAG-tagged transporters, HA-DIRC2, and ATP1A1. The cells were washed three times with PBS. Then, the cells were incubated for 1 h at room temperature in PBS with goat polyclonal anti-mouse IgG antibody coupled to Alexa Fluor Plus 488 (Invitrogen) and goat polyclonal anti-rabbit IgG antibody coupled to Alexa Fluor 594 (Jackson ImmunoResearch) at a dilution of 1:500 and visualized using a confocal laser-scanning microscope (LMS510-META; Zeiss). Caco-2 cells transiently transfected with HA-DIRC2 and cultured in 35-mm glass bottom dishes were similarly treated, using rabbit polyclonal anti-HA antibody and mouse monoclonal anti-ATP1A1 antibody (Abcam) as primary antibodies for detection.

### Uptake study

Uptake assays were done as described previously (Mimura et al, 2017), using transiently transfected COS-7 and HEK293 cells cultured in 24-well coated plates. Briefly, uptake solutions were prepared using Hanks' solution supplemented with 10 mM MES (pH 6.5 and below) or 10 mM Hepes (pH 7.0 and above) and added with [3H]pyridoxine as the substrate. The cells were preincubated for 5 min in 1 ml of substrate-free uptake solution. Uptake assays were initiated by replacing the substrate-free uptake solution with the one containing [3H]pyridoxine (0.25 ml). All procedures were conducted at 37°C. In experiments for intracellular acidification using the protonophores, carbonylcyanide p-trifluoromethoxyphenylhydrazone and carbonylcyanide m-chlorophenylhydrazone were added in the solutions for preincubation and uptake. To examine the effect of ionic conditions, NaCl was replaced as indicated. For inhibition experiments, test compounds were added to the solution only during the uptake period. After termination of [3H] pyridoxine uptake into the cells, the cells were solubilized and the associated radioactivity was determined by liquid scintillation counting. Uptake was normalized to cellular protein content, which was determined by the bicinchoninic acid method (Protein Assay BCA Kit; FUJIFILM Wako Pure Chemical) using BSA as a standard. Uptake assays were similarly conducted using Caco-2 cells cultured in 24-well coated plates.

To examine lysosomal accumulation of pyridoxine in Caco-2 cells, uptake assays were conducted using the cells cultured in six-well coated plates, 4 ml of substrate-free uptake solution for preincubation, and 1 ml of uptake solution containing [3H]pyridoxine. After termination of [3H]pyridoxine uptake into the cells, the lysosomal fraction was isolated for determination of the associated radioactivity and protein content.

### Data analysis

The saturable uptake of pyridoxine by each transporter was analyzed using the Michaelis–Menten model equation as follows: $v = V_{max} \times s / (K_m + s)$. The $V_{max}$ and $K_m$ were estimated by fitting this equation to the experimental profile of the uptake rate ($v$) versus concentration ($s$) of the substrate (pyridoxine) using a nonlinear least-squares regression analysis program, WinNonlin (Certara).

Unless otherwise indicated, the data are presented as the mean ± SD with the number of experiments conducted using different preparations of cells. Each experiment was conducted in duplicate as biological replicates. Statistical analysis was performed using a $t$ test or when multiple comparisons were needed, ANOVA followed by Dunnett's test or Bonferroni test. A level of $P < 0.05$ was considered statistically significant.

## Data Availability

All data are contained within the article.

## Supplementary Information

## Acknowledgements

We thank Hamid M Said, PhD (University of California, Irvine), for helpful discussions and valuable comments in the preparation of the manuscript. This study was supported in part by JPSP KAKENHI Grant Numbers JP18K14953 and JP18KK0453 and the Takeda Science Foundation.

### Author Contributions

S Akino: formal analysis, investigation, visualization, and methodology.
T Yasujima: conceptualization, formal analysis, funding acquisition, investigation, visualization, methodology, and writing—original draft.
T Yamashiro: investigation and methodology.
H Yuasa: conceptualization, supervision, project administration, and writing—review and editing.

### Conflict of Interest Statement

The authors declare that they have no conflict of interest.

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
