## [Reviewer comments · Life Science Alliance]

Life Science Alliance

Disrupted in renal carcinoma 2 (DIRC2/SLC49A4) is a H⁺-driven lysosomal pyridoxine exporter

Shogo Akino, Tomoya Yasujima, Takahiro Yamashiro, and Hiroaki Yuasa

DOI: <https://doi.org/10.26508/lsa.202201629>

Corresponding author(s): Tomoya Yasujima, Nagoya City University

Review Timeline:

Submission Date:	2022-07-25
Editorial Decision:	2022-08-24
Revision Received:	2022-11-10
Editorial Decision:	2022-11-14
Revision Received:	2022-11-20
Accepted:	2022-11-21

Transaction Report:

August 24, 2022

Re: Life Science Alliance manuscript #LSA-2022-01629-T

Dr. Tomoya Yasujima
Nagoya City University
Graduate School of Pharmaceutical Sciences
3-1 Tanabe-Dori
Mizuho-ku
Nagoya, Aichi 4678603
Japan

Dear Dr. Yasujima,

Thank you for submitting your manuscript entitled "Disrupted in renal carcinoma 2 (DIRC2/SLC49A4) is a H⁺-driven lysosomal pyridoxine exporter" to Life Science Alliance. The manuscript was assessed by an expert reviewer, whose comments are appended to this letter. We invite you to submit a revised manuscript addressing the Reviewer comments.

When submitting the revision, please include a letter addressing the reviewer comments point by point.

Thank you for this interesting contribution to Life Science Alliance. We are looking forward to receiving your revised manuscript.

Sincerely,

B. MANUSCRIPT ORGANIZATION AND FORMATTING:

Reviewer #1 (Comments to the Authors (Required)):

The manuscript by Akino et al. characterizes the function of the orphan transporter DIRC2, which was previously shown to be localised in the lysosomal membrane. Though general transport activity of this transporter had been demonstrated in a very unspecific way before, the actual transported substrate has remained unknown to date. Akino et al. now demonstrate that DIRC2 is a H⁺-coupled pyridoxine transporter which they demonstrated in whole-cell uptake assays employing a radiolabelled substrate. These assays are based on re-directing the DIRC2 protein, which is normally localised intracellularly in the lysosomal membrane, to the plasma membrane by mutating a dileucine-based lysosomal transport signal. The authors provide a detailed kinetic analysis of the transport activity and also test several pyridoxine-related molecules for potential transport. Altogether, this characterisation of the DIRC2 function fills an important gap in knowledge as this was not known before. The data in this part of the manuscript is very clear and convincing and, in general, the manuscript is well written.

However, as specified further below it seems to me that the manuscript is a bit blown up with aspects which are either not new (identification of the dileucine motif) or do not add much to the story (Fig. 1). On the other hand the authors dismiss to take the step from identifying the molecular function of DIRC as pyridoxine transporter to put that into a cell biological context. It is a new and by this means exciting finding that lysosomal membranes possess a pyridoxine transporter. However, this raises the question why cells may need such a protein. Having identified the molecular function of DIRC2 should allow to analyse into what happens to pyridoxine trafficking and pyridoxine-dependent reactions when DIRC2 is missing. My feeling is that the following points should be addressed in order to support publication:

Major Points:

The Dileucine motif and the ability to re-direct the protein by mutating this has all been pre-characterized in Savalas et al, 2011 which the authors cite correctly. In this regard, this is totally fine from a formal perspective as it is great to reproduce findings from previous studies. However, for being a simple repetition of something known, this observation consumes in my eyes a bit too much space as it is even listed as an independent bullet point in the synopsis. It leaves a bit impression that the paper is artificially blown-up, which the authors may not wanted to do.

What is the purpose of Figure 1? The authors compare the DIRC2-related family members, but is not clear what this tells us. None of the other proteins has been shown to transport pyridoxine. Two of the proteins were shown to be heme transporters. But the authors do not test if DIRC2 transports heme. On the other hand it is also not tested if any of the others may transport pyridoxine. In my eyes showing this Figure only makes sense if anything is deduced from it or if any experiments formally comparing these different transporters are performed.

Perhaps I overlooked this, but I could not find any hint by the authors how they came about to test DIRC2 for pyridoxine transport activity. At least it seems that this could not be deduced from the bioinformatics analysis in Fig.1 as the most closely related transporters either transport different substrates or it is not known what they transport. Without doubting the data and the conclusions on pyridoxine transport, I think some explanation how the authors find out about this would be nice. As at least according to my knowledge no pyridoxine transport across lysosomal membranes has been described, it does not appear as an obvious compound to start with.

One aspect that is totally ignored is the observation from Savalas et al., that the DIRC2 protein when reaching the lysosomal membrane is proteolytically processed into two fragments, which also seemed to be the form in which the endogenous DIRC2 protein exists. In this previous study, re-directing the protein to the plasma membrane reduced proteolytic cleavage. A central question is if proteolytic cleavage affects transport activity. As a start, the authors should analyse if proteolysis in their experimental systems also occurs. To make it transparent to the reader, DIRC2 blots should not be cropped but shown in full so that amounts of full length and cleaved forms of the protein can be appreciated what ever is present. Beyond that, It would be great if they could find a way to analyse if proteolytic cleavage affects transport. In case that the surface-directed DIRC2 is not cleaved in their set-up, they could try to pre-incubate the cells with recombinant cathepsins at acidic pH and analyse if that induces cleavage of DIRC2.

I think the authors overemphasize a potential link between DIRC2 and renal carcinoma. It is true that the name of the DIRC2

gene (and so also of the protein) comes from being identified at a breakpoint of a chromosomal translocation found in some patients with renal carcinoma. But this is a single report and the functional importance of this translocation for carcinogenesis has to my knowledge not been shown. Therefore, it is very uncertain if DIRC2 is really relevant for carcinogenesis. Therefore, instead of trying to link DIRC2, pyridoxine and cancer in their discussion, the authors should put much more emphasis on the physiological function of DIRC2. DIRC2 is in general a rather broadly expressed protein as the authors demonstrates themselves. I think this leads to the interesting question why and under which conditions lysosomes need a pyridoxine transporter.

In line with the previous comment, the author should invest more to analyse what happens if DIRC2 is missing. So far they do this with siRNA. Generation of a KO cell line should be performed as even low amounts of lysosomal proteins can be sufficient to prevent a phenotype. Is there storage of pyridoxine in lysosomes of such a cell line? The attempts with differential permeabilisation are not a very clean approach. Lysosomes can easily be enriched by subcellular fractionation. Pyridoxine content in such enriched lysosomes from wild type and DIRC2 KO cells should be compared. If no lysosomal pyridoxine storage would be detected, this could also be an important answer as it may indicate that alternative transporters can take over. The authors should also test if there is an impairment of pyridoxine-requiring reactions in the KO cells und basal but also under starvation conditions. It would be interesting to test if these cells may cope less well with situation when no pyridoxine from the outside is supplied.

Minor points:

The quality of the immunofluorescence pictures could be improved.

I am unsure what the experiment in Fig. 4a tells us as the authors look at total cellular levels. DIRC2 may be expected to sequester pyridoxine in the lysosome. But lysosomes are still part of the cell. Why does DIRC2 lead to a reduction of total cellular pyridoxine levels when co-transfected with THTR2 and what do we learn from this?

In the Figure legends the authors specify at several points the number of biological replicates. However, it does not fully become clear if these were derived from independent experiments or not. Perhaps a little more detailed information allowing to judge how often experiments were repeated would be helpful for the reader.

Response to Reviewer #1

We greatly appreciate your kind comments on our manuscript. Taking them into account, we have made revisions to the manuscript. Described below are our answers to your comments, and revisions are marked in red in the revised manuscript.

Major Points:

Comment 1: The Dileucine motif and the ability to re-direct the protein by mutating this has all been pre-characterized in Savalas et al, 2011 which the authors cite correctly. In this regard, this is totally fine from a formal perspective as it is great to reproduce findings from previous studies. However, for being a simple repetition of something known, this observation consumes in my eyes a bit too much space as it is even listed as an independent bullet point in the synopsis. It leaves a bit impression that the paper is artificially blown-up, which the authors may not wanted to do.

Answer: We have deleted the first item in the Synopsis on p. 3, as advised.

Comment 2: What is the purpose of Figure 1? The authors compare the DIRC2-related family members, but is not clear what this tells us. None of the other proteins has been shown to transport pyridoxine. Two of the proteins were shown to be heme transporters. But the authors do not test if DIRC2 transports heme. On the other hand it is also not tested if any of the others may transport pyridoxine. In my eyes showing this Figure only makes sense if anything is deduced from it or if any experiments formally comparing these different transporters are performed.

Answer: We have deleted Figure 1 and the first part titled "Bioinformatic analysis of DIRC2" (p. 6), which mostly referred to the figure in the original manuscript, as advised. In conjunction with this, we have added a sentence in line 4 - 6 from the bottom on p. 5 to provide some more information about the dileucine motif, which was mentioned in the deleted part. We have included the GenBank accession number for DIRC2 cDNA, which was also noted in the deleted part, in the first sentence in the part of "Preparation of plasmids" on p. 14. All the remaining figures have been renumbered.

Comment 3: Perhaps I overlooked this, but I could not find any hint by the authors how they came about to test DIRC2 for pyridoxine transport activity. At least it seems that this could not be deduced from the bioinformatics analysis in Fig.1 as the most closely related transporters either transport different substrates or it is not known what they transport. Without doubting the data and the conclusions on pyridoxine transport, I think some explanation how the authors find out about this would be nice. As at least according to my knowledge no pyridoxine transport across lysosomal membranes has been described, it does not appear as an obvious compound to start with.

Answer: There were no definite reasons to expect pyridoxine transport activity for DIRC2. DIRC2 was identified as a lysosomal pyridoxine transporter in our rather broadly conducted search efforts to identify pyridoxine transporters at plasma and lysosomal membranes. We have revised the initial part in the second paragraph on p. 5 to mention that.

Comment 4: One aspect that is totally ignored is the observation from Savalas et al., that the DIRC2 protein when reaching the lysosomal membrane is proteolytically processed into two fragments, which also seemed to be the form in which the endogenous DIRC2 protein exists. In this previous study, re-directing the protein to the plasma membrane reduced proteolytic cleavage. A central question is if proteolytic cleavage affects transport activity. As a start, the authors should analyze if proteolysis in their experimental systems also occurs. To make it transparent to the reader, DIRC2 blots should not be cropped but shown in full so that amounts of full length and cleaved forms of the protein can be appreciated what ever is present. Beyond that, it would be great if they could find a way to analyze if proteolytic cleavage affects transport. In case that the surface-directed DIRC2 is not cleaved in their set-up, they could try to pre-incubate the cells with recombinant cathepsins at acidic pH and analyze if that induces cleavage of DIRC2.

Answer: We have replaced the western blots with uncropped ones (renumbered Figs. 1B, 3B, and 3F). As shown in the blots, plasma membrane-directed DIRC2-AA was also mostly present as the cleaved product in our study. We have additionally commented on this issue in line 22 on p. 6 - line 2 on p. 7, and also on p. 11 (line 12 - 13), and p. 12 (line 3 - 5).

Comment 5: I think the authors overemphasize a potential link between DIRC2 and renal carcinoma. It is true that the name of the DIRC2 gene (and so also of the protein) comes from being identified at a breakpoint of a chromosomal translocation found in some patients with renal carcinoma. But this is a single report and the functional importance of this

translocation for carcinogenesis has to my knowledge not been shown. Therefore, it is very uncertain if DIRC2 is really relevant for carcinogenesis. Therefore, instead of trying to link DIRC2, pyridoxine and cancer in their discussion, the authors should put much more emphasis on the physiological function of DIRC2. DIRC2 is in general a rather broadly expressed protein as the authors demonstrates themselves. I think this leads to the interesting question why and under which conditions lysosomes need a pyridoxine transporter.

In line with the previous comment, the author should invest more to analyse what happens if DIRC2 is missing. So far they do this with siRNA. Generation of a KO cell line should be performed as even low amounts of lysosomal proteins can be sufficient to prevent a phenotype. Is there storage of pyridoxine in lysosomes of such a cell line? The attempts with differential permeabilization are not a very clean approach. Lysosomes can easily be enriched by subcellular fractionation. Pyridoxine content in such enriched lysosomes from wild type and DIRC2 KO cells should be compared. If no lysosomal pyridoxine storage would be detected, this could also be an important answer as it may indicate that alternative transporters can take over. The authors should also test if there is an impairment of pyridoxine-requiring reactions in the KO cells und basal but also under starvation conditions. It would be interesting to test if these cells may cope less well with situation when no pyridoxine from the outside is supplied.

Answer: Although the role of the lysosomal pyridoxine storage system, including the involvement of DIRC2, in the physiological processes that require pyridoxine remains to be verified, the newly identified function of DIRC2 as a lysosomal pyridoxine exporter should help guiding future studies for verification of the role and exploration of its clinical relevance. We have revised the section of “Concluding remarks” to include this and also to make comments related to the potential link of DIRC2 to carcinogenesis more concise.

As a better approach alternative to the evaluation of pyridoxine accumulation in the intracellular compartment in Caco-2 cells permeabilized by a digitonin treatment, we attempted to demonstrate lysosomal accumulation of pyridoxine by fractionating lysosomes after incubation of Caco-2 cells for pyridoxine accumulation. We could observe lysosomal pyridoxine accumulation and its increase by DIRC2 knockdown. We have added the data as Fig. 4C with Fig. 4D, which shows western blots to confirm successful fractionation of lysosomes, and described the results in line 16 - 21 on p. 12. Panels in Fig. 4 were relabeled accordingly. Although the changes in cellular and lysosomal accumulation were modest and it may be because of the activity of residual DIRC2 in DIRC2-KD cells, the results support our suggested role of DIRC2 as a lysosomal pyridoxine exporter. The handling of Caco-2 cells and related methods for this set of experiments have been additionally described in line 25 on p. 15 - line 2 on p. 16, line 14 - 16 and line 24 - 26 on p. 17, and the second paragraph on p. 19. The method for isolation of lysosomal fraction has been additionally described on p. 16. We have deleted the data obtained using digitonin-treated, permeabilized cells (Fig. 5G in the original manuscript), a paragraph mostly referring to the data (last paragraph on p. 13 in the original), and a paragraph for the method (second paragraph on p. 19).

We would greatly appreciate if you could kindly understand these handlings regarding the issues raised here.

Minor points:

Comment 6: The quality of the immunofluorescence pictures could be improved.

Answer: We have replaced the images in Fig. 4B, which may have been low in quality, with improved ones.

Comment 7: I am unsure what the experiment in Fig. 4a tells us as the authors look at total cellular levels. DIRC2 may be expected to sequester pyridoxine in the lysosome. But lysosomes are still part of the cell. Why does DIRC2 lead to a reduction of total cellular pyridoxine levels when co-transfected with THTR2 and what do we learn from this?

Answer: When pyridoxine is supplied from the extracellular medium, the extent of lysosomal accumulation would affect the total cellular level of pyridoxine. In this situation, an enhancement of lysosomal export would lead to a decrease in lysosomal accumulation and, accordingly, a decrease in the total cellular level. Therefore, the decrease in the total cellular level in the presence of FLAG-THTR2 can be interpreted as a result of an increase in the lysosomal export by the introduction of HA-DIRC2, in accordance with its suggested role as a lysosomal pyridoxine exporter, as described in line 10 - 12 on p. 11. In the absence of FLAG-THTR2, cellular pyridoxine uptake was presumably too low for a change in the total cellular accumulation caused by a HA-DIRC2-induced change in lysosomal accumulation to be detectable, as described in line 1 - 4 from the bottom on p. 10.

Comment 8: In the Figure legends the authors specify at several points the number of biological replicates. However, it does not fully become clear if these were derived from independent experiments or not. Perhaps a little more detailed information allowing to judge how often experiments were repeated would be helpful for the reader.

Answer: Those experiments were replicated using different preparation of cells. We have additionally noted that at every appearance in legends.

In addition to revisions for specific points, the length indicated for the scale bar was corrected to be 10 μm in the legends for Figs. 1A, 3C, and 4B. Some other minor editorial corrections have also been made in the revised manuscript. They are also marked in red.

November 14, 2022

RE: Life Science Alliance Manuscript #LSA-2022-01629-TR

Dr. Tomoya Yasujima
Nagoya City University
Graduate School of Pharmaceutical Sciences
3-1 Tanabe-dori
Mizuho-ku
Nagoya, Aichi 4678603
Japan

Dear Dr. Yasujima,

Thank you for submitting your revised manuscript entitled "Disrupted in renal carcinoma 2 (DIRC2/SLC49A4) is a H⁺-driven lysosomal pyridoxine exporter". We would be happy to publish your paper in Life Science Alliance pending final revisions necessary to meet our formatting guidelines.

- please add a Running Title, Abstract, and an alternate abstract to our system
- please add the Twitter handle of your host institute/organization as well as your own or/and one of the authors in our system
- please consult our manuscript preparation guidelines <https://www.life-science-alliance.org/manuscript-prep> and make sure your manuscript sections are in the correct order
- the file you labeled as "Cover Art" can be uploaded as a Graphical Abstract, if you wish
- In the Figure 4 legend, panel I is described, but there is no panel I in the figure, and there is not a callout for this panel in the text

A. FINAL FILES:

B. MANUSCRIPT ORGANIZATION AND FORMATTING:

Sincerely,

November 21, 2022

RE: Life Science Alliance Manuscript #LSA-2022-01629-TRR

Dr. Tomoya Yasujima
Nagoya City University
Graduate School of Pharmaceutical Sciences
3-1 Tanabe-dori
Mizuho-ku
Nagoya, Aichi 4678603
Japan

Dear Dr. Yasujima,

Thank you for submitting your Research Article entitled "Disrupted in renal carcinoma 2 (DIRC2/SLC49A4) is a H⁺-driven lysosomal pyridoxine exporter". It is a pleasure to let you know that your manuscript is now accepted for publication in Life Science Alliance. Congratulations on this interesting work.

DISTRIBUTION OF MATERIALS:

Again, congratulations on a very nice paper. I hope you found the review process to be constructive and are pleased with how the manuscript was handled editorially. We look forward to future exciting submissions from your lab.

Sincerely,
